# Clinical Evaluation of the BIOFIRE SPOTFIRE Respiratory Panel

**DOI:** 10.3390/v16040600

**Published:** 2024-04-13

**Authors:** Wai-Sing Chan, Christy Wing-Yiu Ho, Tsz-Ching Chan, Jeffrey Hung, Man-Yan To, Sau-Man Leung, Ka-Chun Lai, Ching-Yan Wong, Chin-Pang Leung, Chun-Hang Au, Thomas Shek-Kong Wan, Jonpaul Sze-Tsing Zee, Edmond Shiu-Kwan Ma, Bone Siu-Fai Tang

**Affiliations:** Department of Pathology, Hong Kong Sanatorium & Hospital, Hong Kong SAR, China; waising.chan@connect.polyu.hk (W.-S.C.); tommy.ch.au@hksh.com (C.-H.A.);

**Keywords:** BIOFIRE, FilmArray, flu, PCR, respiratory, SARS-CoV-2, SPOTFIRE, TORCH

## Abstract

The BIOFIRE SPOTFIRE Respiratory (R) Panel is a novel, in vitro diagnostic PCR assay with 15 pathogen targets. The runtime is about 15 min which is the shortest among similar panels in the market. We evaluated the performance of the SPOTFIRE R Panel with 151 specimens, including 133 collected from the upper respiratory tract (URT), 13 from the lower respiratory tract (LRT) and 5 external quality assessment program (EQAP) samples. The respiratory specimens were enrolled throughout the first two post-COVID-19 influenza seasons in Hong Kong (March to December 2023). For URT specimens, full concordance was observed between the SPOTFIRE R Panel and the standard-of-care FilmArray Respiratory 2.1 *plus* Panel (RP2.1*plus*) for 109 specimens (109/133, 81.95%). After discrepant analysis, the SPOTFIRE R Panel identified more pathogens than the RP2.1*plus* in 15 specimens and vice versa in 3 specimens. The per-target negative and positive percentage agreement (NPA and PPA) were 92.86–100% except the PPA of adenovirus (88.24%). For LRT and EQAP samples, all results were fully concordant. To conclude, the performance of the SPOTFIRE R Panel was comparable to the RP2.1*plus*.

## 1. Introduction

Respiratory infection is among the leading causes of disease burden and mortality worldwide [1,2,3]. More than a health crisis, the socioeconomic impact brought by a highly transmissible pathogenic respiratory virus can be catastrophic, which is evident by the recent coronavirus disease 2019 (COVID-19) pandemic [4]. The initial clinical manifestations of upper respiratory tract infection are usually indistinguishable among different etiologies. Albeit most of the symptoms are self-limiting for healthy adults, the sequelae can be serious for children, the elderly and the immunocompromised [5,6,7]. Rapid identification of the culprit may guide optimal patient management and timely infection control measures.

Recent advances in molecular methods have revolutionized infectious disease diagnostics. The paradigm has been shifting from tedious culture-based methods to rapid nucleic acid amplification-based testing. For instance, there are several commercial in vitro diagnostic assays in the market, which can detect more than 20 respiratory pathogen targets in 45 to 90 min [8,9,10]. Recently, the BIOFIRE SPOTFIRE Respiratory (R) Panel has been cleared by the U.S. Food and Drug Administration (FDA) 510(k) and waived by the Clinical Laboratory Improvement Amendments [11,12]. Its principle of operation is very similar to that of the FilmArray Respiratory Panel from the same company, which automates and integrates quality-controlled nucleic acid extraction, reverse transcription, nested amplification of pathogen targets, fluorescent signal detection and melt curve analysis of amplified targets in a closed, disposable pouch, with data interpreted automatically. A comparison between the SPOTFIRE R Panel and the latest version of FilmArray Respiratory Panel (RP2.1*plus*) is shown in Figure 1. The SPOTFIRE R Panel is compatible with the novel SPOTFIRE System, whereas the RP2.1*plus* is run on FilmArray 2.0 or TORCH. The runtime of the SPOTFIRE R Panel is about 15 min which is one-third of the RP2.1*plus*. The pathogen targets of the SPOTFIRE R Panel are the same as those of the RP2.1*plus* except the Middle East respiratory syndrome coronavirus (MERS-CoV), and the subtype results of the seasonal coronavirus (229E, NL63, HKU1 and OC43) and parainfluenza virus (type 1 to 4) are not available.

In the literature, there are limited evaluation data on the clinical performance of the SPOTFIRE R Panel. In the light of this, we evaluated the performance of the SPOTFIRE R Panel in a clinical laboratory setting. The standard-of-care method which uses the RP2.1*plus* was the primary comparator.

## 2. Materials and Methods

### 2.1. Specimens

A total of 151 specimens were enrolled in this study, encompassing 133 upper respiratory tract (URT) specimens (swabs of nasal cavity/nasopharynx/throat and posterior oropharyngeal saliva), 13 lower respiratory tract (LRT) specimens (bronchoalveolar lavage fluid, endotracheal aspirate and sputum) and 5 external quality assessment program (EQAP) samples. The human respiratory specimens were referred to the Department of Pathology, Hong Kong Sanatorium & Hospital for standard-of-care testing of respiratory pathogens from March to December 2023, using the RP2.1*plus* on FilmArray 2.0 or TORCH (BIOFIRE Diagnostics, Salt Lake City, UT, USA). Aliquots of residual specimens were kept at 4 °C for 1 month and −80 °C for long-term storage. The specimens were selected to cover all the pathogen targets of the SPOTFIRE R Panel. The RP2.1*plus* quality controls of all enrolled specimens were valid. The minimum sample volume was 0.5 mL.

### 2.2. Testing with SPOTFIRE R Panel

The enrolled specimens were tested with the SPOTFIRE R Panel on the SPOTFIRE System (BIOFIRE Diagnostics, Salt Lake City, UT, USA) from September to December 2023, by 2 operators blinded to the RP2.1*plus* results. Briefly, archived specimens were brought to ambient temperature before testing. The Class II A2 biosafety cabinet work area and the pouch-loading station were cleaned with disinfectant wipes. A SPOTFIRE pouch was removed from the vacuum-sealed package and properly labelled. The pouch, the red-capped sample injection vial and the blue-capped hydration injection vial were inserted into the loading station. The hydration injection vial was unscrewed and inserted into the pouch hydration port to reconstitute the lyophilized reagents. For swabs in viral transport medium, each specimen was thoroughly homogenized and dispensed into the sample injection vial (containing sample buffer) using the transfer pipette provided (approximated 0.3 mL). For posterior oropharyngeal saliva and LRT specimens, 0.5 mL of each specimen was mixed with an equal volume of working Sputasol (Oxoid Ltd., Basingstoke, UK) at ambient temperature, and 0.3 mL of the liquefied specimen was dispensed into the sample injection vial containing sample buffer. After adding the specimen, the cap of sample injection vial was closed tightly and the specimen was mixed with sample buffer by gently inverting 3 times. The sample injection vial was unscrewed after 5 s and inserted into the pouch sample port, and the specimen was pulled into the pouch by vacuum. On the touchscreen of the SPOTFIRE System, an available module was selected. The pouch and specimen barcodes were scanned. The pouch was then inserted into the module with blinking blue light. The pouch was then pulled into the chamber, and the run started automatically.

### 2.3. Discrepant Analysis

The RP2.1*plus* was the primary comparator for evaluating the performance of the SPOTFIRE R Panel. Discrepant results were verified by alternative methods, as shown in Table 1.

### 2.4. Statistical Analysis

Positive and negative percent agreement (PPA and NPA) were calculated by using the following formulae:PPA = TP/(TP + FN) × 100%
NPA = TN/(TN + FP) × 100%

FN, FP, TN and TP stand for the number of false-negative, false-positive, true-negative and true-positive specimens, respectively. The 95% confidence intervals (CI) for PPA and NPA were calculated using the online version of GraphPad software (modified Wald method) [17].

## 3. Results

### 3.1. Patient Demographics

The demographic features are summarized in Appendix A. The patients were divided into six age ranges according to the U.S. National Institutes of Health (NIH) guidelines [18]. Children comprised the majority (89/146, 60.96%), followed by adults (25/146, 17.12%) and older adults (19/146, 13.01%). The pattern was similar to the patient population tested with the RP2.1*plus* in 2022 and the first quarter of 2023 [19]. Male patients (82/146, 56.16%) or inpatients (76/146, 52.05%) were slightly more in number than female (64/146, 43.84%) or outpatients (70/146, 47.95%), respectively.

### 3.2. Performance of the SPOTFIRE R Panel

A total of 153 runs were performed for 151 specimens, and the results of 2 were invalid due to internal process control failure (2/153, 1.31%). The repeated runs were valid.

Table 2 summarizes the frequency of pathogen detection for the URT specimens. The overall positive rate of the SPOTFIRE R and RP2.1*plus* were the same (121/133, 90.98%), and full concordance was observed for 109 specimens (81.95%). From the raw data, the SPOTFIRE R Panel detected more pathogens than the RP2.1*plus* in 18 specimens and vice versa in 5 specimens.

Appendix A summarizes the detection frequency of each pathogen target and the results of discrepant analysis. No discrepancy was observed for the detection of metapneumovirus (6/6), influenza A virus (41/41), influenza B virus (8/8), parainfluenza virus 1–3 (19/19), *Bordetella pertussis* (2/2), *Chlamydia pneumoniae* (1/1) and *Mycoplasma pneumoniae* (13/13).

Considering the raw data, there were 30 discrepant results between the SPOTFIRE R Panel and RP2.1*plus*, with 24 ‘false positives’ and 6 ‘false negatives’ for the former. Results of alternative methods were concordant with 17 ‘false positives’ and 2 ‘false negatives’. The 17 resolved false positives were rhinovirus/enterovirus (*n* = 11), adenovirus (*n* = 2), SARS-CoV-2 (*n* = 2), influenza A virus H3 (*n* = 1) and parainfluenza virus 4 (*n* = 1). The two resolved false negatives were SARS-CoV-2 (*n* = 1) and respiratory syncytial virus (n = 1).

Appendix A summarizes the results of discrepant analysis for co-detections. From the raw data, the SPOTFIRE R Panel detected more pathogens than the RP2.1*plus* in 18 specimens and vice versa for 5 specimens. After discrepant analysis, the SPOTFIRE R Panel detected more pathogens than the RP2.1*plus* in 15 specimens and vice versa for 3 specimens.

Table 3 summarizes the per-target performance of the SPOTFIRE R Panel for the URT specimens after resolution of discrepancies. The PPAs ranged from 88.24 to 100% and NPAs ranged from 97.41 to 100%.

For both the LRT specimens and EQAP samples, the results of the SPOTFIRE R Panel were fully concordant with the RP2.1*plus* (Appendix A).

## 4. Discussion

From the late COVID-19 pandemic, a rebound in the activities of non-SARS-CoV-2 respiratory pathogens has been observed worldwide [19,20,21,22,23,24]. In clinical laboratories, the attention and resources have been shifting from SARS-CoV-2 detection to multiplex detection of SARS-CoV-2 plus other respiratory pathogens. At the time of writing, Hong Kong is experiencing the third influenza season since the end of the COVID-19 pandemic. The first post-COVID-19 influenza season lasted from April to May, and the second one lasted from August to November in 2023 [25]. Compared with 2022, the average number of multiplex testings, the positive rate and the diversity of pathogens detected in our laboratory rose two- to four-fold in 2023. It can be speculated that a two-third reduction of the existing assay runtime may help alleviate the increasing test burden and streamline laboratory operations, and this warrants further cost-effectiveness analysis.

We enrolled respiratory specimens throughout the first two post-COVID-19 influenza seasons, for the sake of including the latest circulating strains. Owing to the scope of study, we did not perform strain characterization, albeit further genomic studies could be of epidemiological interest.

There are limited evaluation data on the clinical performance of the SPOTFIRE R Panel in the literature. The most comprehensive study was pursued by the manufacturer for substantial equivalence determination to other FDA-cleared comparator methods [26]. A combination of prospectively collected and archived nasopharyngeal swabs (NPS) were used. The prospective study was conducted at four U.S. and one ex-U.S. clinical setting from December 2020 to June 2021. A total of 1120 NPS were enrolled, and the results were compared to a combination of two FDA-cleared panels. The PPA and NPA were 96.3–100% and 90.6–100%, respectively. On the other hand, 542 archived NPS from various clinical laboratories around the world were tested at the four U.S. sites. The PPA and NPA were 96.0–100% and 96.7–100%, respectively. The NPA of rhinovirus/enterovirus (RV/EV) for prospectively collected NPS was the lowest (695/767, 90.6%). Discrepant analysis confirmed the presence of RV/EV in 54 out of 72 discordant specimens (75%), and the NPA was 97.65% (749/767) after discrepant analysis. Similarly, the SPOTFIRE R Panel appeared to be more sensitive to rhinovirus in our study. Rhinovirus was identified by the SPOTFIRE R Panel only in 11 specimens and by the RP2.1*plus* only in 1 specimen. Summing up our findings, the NPA and PPA of all pathogen targets were above 90% except the PPA of adenovirus (15/17, 88.24%). Discrepant analysis revealed that the Ct values of adenovirus were 40 in the two false-negative specimens. The PPA of adenovirus was 97% (prospectively collected NPS, 32/33) and 100% (archived NPS, 31/31) in the study by the manufacturer. Stochastic effects on the amplification of low-viral-load specimens and the relatively small number of adenovirus-positive specimens might account for the low PPA of adenovirus in our study.

For the RP2.1*plus*, a prospective clinical evaluation was pursued by the manufacturer to assess the performance for SARS-CoV-2 detection (details from the package insert of the RP2.1*plus*, BFR0000-8307-03 June 2022). The study was conducted at three geographically distinct sites in the U.S. from July to October 2020. A total of 524 NPS were enrolled, and the results were compared with a composite comparator of three tests with FDA emergency use authorization. The PPA and NPA were 98.4% (61/62) and 98.9% (457/462), respectively. For non-SARS-CoV-2 pathogens, the performance of the RP2.1*plus* was compared with its prior version, the RP2*plus*, using 220 archived specimens. The overall PPA and NPA were 97.6% and 99.8%, respectively, excluding MERS-CoV.

Notably, the SPOTFIRE R Panel identified more co-detections than the RP2.1*plus*. The additional targets co-detected were rhinovirus (n = 11), SARS-CoV-2 (n = 2), adenovirus (n = 2) and parainfluenza virus 4 (n = 1). From the literature, the clinical outcome of respiratory co-infection may vary for different pathogen combinations. For instance, Esper and co-authors reported that influenza patients co-infected with rhinovirus tended to have milder disease [27]. On the other hand, co-infection of rhinovirus might be associated with refractory *M. pneumoniae* pneumonia and a higher frequency of *B. pertussis* LRT infection among hospitalized children [28,29]. Furthermore, concomitant infection of *M. pneumoniae* or parainfluenza virus 4 in COVID-19 patients was associated with worse clinical outcomes [30,31]. A more complete picture of microbial profiles with accumulated knowledge on the clinical outcomes of co-infection may aid etiologic diagnosis and prognosis.

In addition to URT specimens, we included 13 LRT specimens for compatibility testing. The results of both assays were fully concordant. A similar study was pursued by Hughes and co-authors to evaluate the performance of the RP2 for viral pathogen detection in LRT specimens, using the BIOFIRE Pneumonia Panel as comparator [32]. A total of 200 LRT specimens were enrolled, encompassing bronchoalveolar lavage fluid, bronchial wash, sputum and tracheal aspirate. The overall PPA and NPA were 87% (71–95%) and 100% (99–100%), respectively. Further studies with larger and more representative sample pools are warranted to thoroughly assess the performance for LRT specimens.

This study was constrained by the small sample size of LRT specimens, negative specimens and specimens positive for rare pathogens, including *Bordetella parapertussis*, *Bordetella pertussis* and *Chlamydia pneumoniae*. An increased sample size might allow for better assessment of assay performance with consideration of local prevalence. For the SPOTFIRE R Panel, the spectrum of testing is narrower than that of the RP2.1*plus*. It does not include MERS-CoV and cannot distinguish between subtypes of seasonal coronavirus and parainfluenza virus. Concerning alternative methods for discrepant analysis, we did not have complete information on the detection targets and performance characteristics for some assay, and with different assay principles it is not easy to compare their sensitivity with the SPOTFIRE R Panel or RP2.1*plus* directly. Similar to other molecular assays, neither the SPOTFIRE R Panel nor RP2.1*plus* provides information on the viability or quantity of the pathogens detected or distinguishing between co-detection and co-infection. In this regard, the valuable data on target Ct values and melt curves may be useful to laboratorians and clinicians for better result interpretation and patient management.

## 5. Conclusion

The performance of the SPOTFIRE R Panel was comparable to that of the RP2.1*plus*.

## Figures and Tables

**Figure 1 viruses-16-00600-f001:**
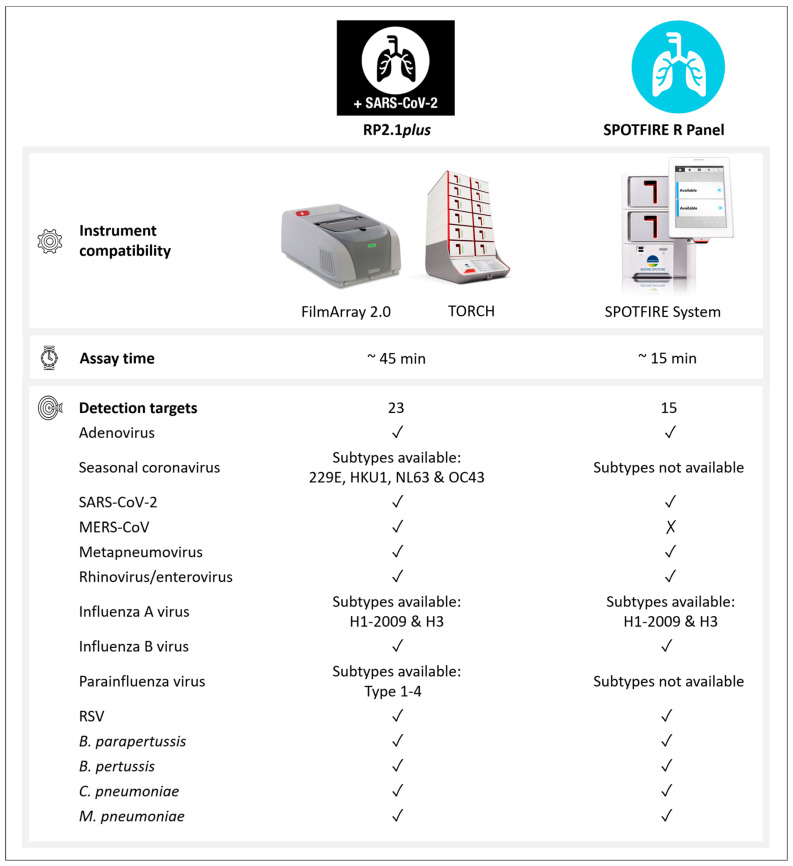
Comparison between FilmArray Respiratory 2.1 *plus* Panel (RP2.1*plus*) and SPOTFIRE Respiratory (R) Panel. The images are acquired from the website of BIOFIRE Diagnostics (Salt Lake City, UT, USA). MERS-CoV, Middle East respiratory syndrome coronavirus; RSV, respiratory syncytial virus; SARS-CoV-2, severe acute respiratory syndrome coronavirus 2; ✓, available; ✗, not available.

**Table 1 viruses-16-00600-t001:** Methods used for discrepant analysis.

Pathogens	Methods	Target Genes	Sensitivity	References
Adenovirus	LightMix Modular Adenovirus (TIB-Molbiol, Berlin, Germany)	Not available	Not available	
*Bordetella* *parapertussis*	Conventional PCR BPpara-1 and 2 primer pair	Insertion sequence IS*1001*	High copy number in *B. parapertussis* and some *B. bronchiseptica*	[13]
Seasonal coronavirus	LightMix Modular panCoronavirus (TIB-Molbiol, Berlin, Germany)	Polyprotein gene	10–100 genome equivalent copies or less per reaction	
Enterovirus	Reverse transcription, semi-nested conventional PCR **First PCR:** primers 012 and 011 **Second PCR:** primers 040 and 011	Capsid protein gene VP1	Not available	[14]
Influenza A virus	LightMix Modular Influenza A H1 (H1N1 sw) and H3 (H3N2) (TIB-Molbiol, Berlin, Germany)	Hemagglutinin gene	Not available	
Parainfluenza virus (PIV)	Reverse transcription, conventional PCR **PIV1:** primers PIV1-F2/R2 **PIV2:** primers PIV2-F2/R2a/R2b **PIV3:** primers PIV3-F2/R2 **PIV4:** primers PIV4-F1/R1	**PIV1-3:** hemagglutinin-neuraminidase gene **PIV4:** nucleocapsid gene	Not available	[15]
Respiratory syncytial virus	Xpert Xpress CoV-2/Flu/RSV *plus* (Cepheid, Sunnyvale, CA, USA)	Nucleocapsid gene	0.33–0.37 TCID_50_/mL	
Rhinovirus	Reverse transcription, semi-nested conventional PCR **First PCR:** primers P1-1 and P3-1 **Second PCR:** primers P1-1 and P2-1/2/3	5′ non-coding region	Not available	[16]
SARS-CoV-2	Xpert Xpress CoV-2/Flu/RSV *plus* (Cepheid, Sunnyvale, CA, USA)	Nucleocapsid gene Envelope gene RNA-dependent RNA polymerase gene	138 copies/mL	
	Xpert Xpress SARS-CoV-2 (Cepheid, Sunnyvale, CA, USA)	Nucleocapsid gene Envelope gene	0.0200 PFU/mL	

**Table 2 viruses-16-00600-t002:** Frequency of pathogen detection for upper respiratory tract specimens.

	No. of Positive Specimens		Remarks
	RP2.1*plus*	SPOTFIRE R	
**Detected**	**121**		^a^ SPOTFIRE R Panel detected more pathogens than RP2.1*plus* for 12 specimens. For 1 specimen, the result of RP2.1*plus* was ‘influenza A H1-2009’ and that of SPOTFIRE R Panel was ‘influenza A no subtype’. The result was classified as partial concordance with the same number of pathogens detected. ^b^ SPOTFIRE R Panel detected more pathogens than RP2.1*plus* for 5 specimens, and vice versa. ^c^ Coronavirus NL63, parainfluenza virus 2 and 4 were detected by RP2.1*plus* while seasonal CoV, PIV and rhinovirus/enterovirus were detected by SPOTFIRE R Panel. SPOTFIRE R Panel detected more pathogen types than RP2.1*plus* albeit the number of positive calls were the same. ^d^ RP2.1*plus* detected more pathogens than SPOTFIRE R Panel for 5 specimens. For 1 specimen, the result of RP2.1*plus* was ‘influenza A H1-2009’ and that of SPOTFIRE R Panel was ‘influenza A no subtype’. The result was classified as partial concordance with the same number of pathogens detected. ^e^ SPOTFIRE R Panel detected more pathogens than RP2.1*plus* for all 18 specimens. For details of each pathogen target, please refer to Appendix A.
*Full concordance*	*97*		
*Partial concordance*	*24*		
**For partial concordance:**			
1 pathogen	13 ^a^	6 ^d^	
2 pathogens	10 ^b^	10 ^e^	
3 pathogens	1 ^c^	6 ^e^	
4 pathogens	0	1 ^e^	
5 pathogens	0	1 ^e^	

**Not detected**	**12**		**From the raw data, SPOTFIRE R Panel detected more pathogens than RP2.1*plus* in 18 specimens, and vice versa in 5 specimens.**
*Full concordance*	*12*		

**Table 3 viruses-16-00600-t003:** Per-target performance of the SPOTFIRE R Panel for upper respiratory tract specimens.

Pathogen Targets	FN	FP	TN	TP	PPA	95% CI for PPA (%)	NPA	95% CI for NPA (%)
Adenovirus	2	3	113	15	88.24	64.41 to 97.97	97.41	92.34 to 99.45
Coronavirus (seasonal)	0	1	114	18	100	79.33 to 100	99.13	94.76 to >99.99
SARS-CoV-2	0	0	118	15	100	76.14 to 100	100	96.21 to 100
Metapneumovirus	0	0	127	6	100	55.72 to 100	100	96.47 to 100
Rhinovirus/enterovirus	1	1	98	33	97.06	83.78 to >99.99	98.99	93.95 to >99.99
Influenza A virus	0	0	92	41	100	89.79 to 100	100	95.19 to 100
Influenza A virus/H1-2009	1	0	119	13	92.86	66.46 to >99.99	100	96.24 to 100
Influenza A virus A/H3	0	0	106	27	100	85.24 to 100	100	95.80 to 100
Influenza B virus	0	0	125	8	100	62.78 to 100	100	96.42 to 100
Parainfluenza virus	0	0	110	23	100	83.09 to 100	100	95.95 to 100
Respiratory syncytial virus	0	1	123	9	100	65.54 to 100	99.19	95.13 to >99.99
*Bordetella parapertussis*	0	1	129	3	100	38.25 to 100	99.23	95.34 to >99.99
*Bordetella pertussis*	0	0	131	2	100	29.02 to 100	100	96.58 to 100
*Chlamydia pneumoniae*	0	0	132	1	100	16.75 to 100	100	96.60 to 100
*Mycoplasma pneumoniae*	0	0	120	13	100	73.41 to 100	100	96.27 to 100

CI, confidence interval; FN, false negative; FP, false positive; NPA, negative percent agreement; PPA, positive percent agreement; TN, true negative; TP, true positive.

## Data Availability

The data presented in this study are available on request from the corresponding author.

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
