# Peer review of "Clinical Evaluation of the BIOFIRE SPOTFIRE Respiratory Panel"

_viruses, 2024, doi:10.3390/v16040600_

Round 1

Reviewer 1 Report

Comments and Suggestions for Authors

The authors compared the clinical performance of the The BIOFIRE SPOTFIRE Respiratory (R) Panel to that of the standard-of-care FilmArray Respiratory 2.1 plus Panel (RP2.1plus). the data are sound and support the manuscript conclusions. However, the ethics statement and approval from an ethic committee was not included in the manuscript.

Reviewer 2 Report

Comments and Suggestions for Authors

Suggestions and comments

Manuscript ID: viruses-2909930

Type: Communication

Title: Clinical evaluation of the BIOFIRE SPOTFIRE Respiratory Panel

Summary section:

1.        The summary must be completely restructured, optimization and timely attention are mentioned but this has not been measured according to the results. Since only performance is analyzed. There is no correlation of the positives according to clinical symptoms, nor indicators of timely care such as the time until receiving treatment according to the intervention group or the control group.

Introduction section

2.         A paragraph must be incorporated that describes studies with results from both panels

Materials and methods section

3.        The collection of samples has not been homogeneous; it is important in the validation, comparison or correlation processes of tests that all samples are taken in the same way. There is a great difference between samples taken from the upper airways vs. the lower airways at the detection limit due to this, for example, the Film array panel asks you to choose which protocol to follow according to the type of sample. This is not indicated in the described methodology and is analyzed as a whole.

4. The methodological basis of the SPOTFIRE system (BIOFIRE Diagnostics, Salt 

Lake City, UT, USA) is not mentioned.

5. It is not mentioned because the population included both neonates and the elderly. Justify this.

Results Section

6. Table 1 can be eliminated since it contributes little to the results. Perhaps it is best described as a short paragraph.

7. Table 3 is duplicate and both contain different data

Conclusion section:

9. It must be linked to results. Execution time has not been evaluated.

10. In general, the manuscript has to be refocused towards the knowledge of local molecular epidemiology with these results that compare in a very limited way 2 multipathogen detection panels. No analyzes have been carried out to optimize the tests, nor have indicators of timely care been developed regarding the use of one test or another. For this reason this should not go in the manuscript.

Another limitation is that the clinical characteristics of the patients who have tested positive have not been considered, no less in the monoinfected or coinfected group according to the results of the panels.

Comments on the Quality of English Language

 Minor editing of English language required 

Reviewer 3 Report

Comments and Suggestions for Authors

1) BIOFIRE SPOTFIRE R Panel  demonstrates greater efficacy within a shorter time frame compared to RP2.1plus. The advantage lies in its ability to showcase performance. However, a limitation exists in the inability to distinguish between subtypes of certain viruses.

2) BIOFIRE SPOTFIRE R Panel is an excellent tool for rapid identification. However, in comparison to general procedures that do not necessitate rapid identification, a consideration of cost-effectiveness should be provided.

3) Various molecular tests are employed to examine the resulting discrepancies. The method priciple applied is required to determine whether the performance is superior to both BIOFIRE SPOTFIRE R Panel and RP2.1plus or guaranteed.

4) It is advisable to create a schematic representation of the distinctions between the BIOFIRE SPOTFIRE R Panel and RP2.1plus for better clarity.

5) Tables 3, 4, and 5 are suitable for inclusion as supplementary files.

Round 2

Reviewer 2 Report

Comments and Suggestions for Authors

No further comments

Comments on the Quality of English Language

 Minor editing of English language required

Reviewer 3 Report

Comments and Suggestions for Authors

.